# Effects of Allelochemicals, Soil Enzyme Activities, and Environmental Factors on Rhizosphere Soil Microbial Community of *Stellera chamaejasme* L. along a Growth-Coverage Gradient

**DOI:** 10.3390/microorganisms10010158

**Published:** 2022-01-12

**Authors:** Jinan Cheng, Hui Jin, Jinlin Zhang, Zhongxiang Xu, Xiaoyan Yang, Haoyue Liu, Xinxin Xu, Deng Min, Dengxue Lu, Bo Qin

**Affiliations:** 1Key Laboratory of Chemistry of Northwestern Plant Resources of Chinese Academy of Sciences/Key Laboratory for Natural Medicines of Gansu Province, Lanzhou Institute of Chemical Physics, Chinese Academy of Sciences, Lanzhou 730000, China; chengjn2019@lzu.edu.cn (J.C.); ruoxi020xw@163.com (X.Y.); liuhaoyue0820@163.com (H.L.); XUXX@licp.cas.cn (X.X.); mindeng@licp.cas.cn (D.M.); 2Center of Grassland Microbiome, State Key Laboratory of Grassland Agro-Ecosystems, Lanzhou University, Lanzhou 730000, China; jlzhang@lzu.edu.cn; 3Animal, Plant & Food Inspection Center of Nanjing Customs, Nanjing 210000, China; konphy@126.com; 4Institute of Biology, Gansu Academy of Sciences, Lanzhou 730000, China; konphpy@126.com

**Keywords:** allelochemicals, microbial community, environmental factors, interaction, *S. chamaejasme*

## Abstract

Allelochemicals released from the root of *Stellera chamaejasme* L. into rhizosphere soil are an important factor for its invasion of natural grasslands. The aim of this study is to explore the interactions among allelochemicals, soil physicochemical properties, soil enzyme activities, and the rhizosphere soil microbial communities of *S. chamaejasme* along a growth-coverage gradient. High-throughput sequencing was used to determine the microbial composition of the rhizosphere soil sample, and high-performance liquid chromatography was used to detect allelopathic substances. The main fungal phyla in the rhizosphere soil with a growth coverage of 0% was Basidiomycetes, and the other sample plots were Ascomycetes. Proteobacteria and Acidobacteria were the dominant bacterial phyla in all sites. RDA analysis showed that neochamaejasmin B, chamaechromone, and dihydrodaphnetin B were positively correlated with Ascomycota and Glomeromycota and negatively correlated with Basidiomycota. Neochamaejasmin B and chamaechromone were positively correlated with Proteobacteria and Actinobacteria and negatively correlated with Acidobacteria and Planctomycetes. Allelochemicals, soil physicochemical properties, and enzyme activity affected the composition and diversity of the rhizosphere soil microbial community to some extent. When the growth coverage of *S. chamaejasme* reached the primary stage, it had the greatest impact on soil physicochemical properties and enzyme activities.

## 1. Introduction

The rhizosphere refers to the part of the soil mediated by microorganisms and affected by the root system, which is directly affected by root exudates and related soil microorganisms [1]. Rhizosphere soil is important not only for plant nutrition, health, and quality, but also for microbial-driven carbon sequestration, ecosystem function, and nutrient cycling in terrestrial ecosystems [2]. It is also the most important place for rhizosphere microorganisms to multiply and grow [3]. Soil microorganisms play an important role in improving soil physical and chemical properties, regulating soil microbial community and diversity, and maintaining soil quality and fertility [4].

Some studies have shown that one third of plant photosynthate is released into soil in the form of root exudates, which can affect the rhizosphere microbial community [5]. Allelopathy, as a new weapon of species invasion, has gained a prominent position in the field of invasive biology [6]. The chemical substances produced by plants affect neighboring plants and soil microorganisms, ultimately changing the structure of plant communities [7]. Currently, scholars are focusing more and more on the allelopathic interaction between plants and their rhizosphere soil, but there are still few related studies. Studies have shown that allelochemicals released by plant roots play a vital role in interactions between soil microorganisms [8]. Adding rice allelochemicals to the soil will stimulate or inhibit certain microbial populations in the soil, thereby affecting the release of allelochemicals and the soil nutrient composition [9]. Brazilian pepper trees use allelochemicals to manipulate the soil fungal and bacterial community structures and recruit AMF fungi to enhance nutrient uptake, drought resistance, and disease resistance, as well as to destroy local soil microbial communities [10]. *Alliaria fololata* is a cruciferous plant that successfully destroys the beneficial fungi that are symbiotic with local trees through isothiocyanate (ITC) allelopathy [11]. In addition, ITC also had a significant impact on the rhizosphere soil bacterial community of *Arabidopsis thaliana* [12]. Root exudates were growth regulators in the process of peanut–soil feedback. Phenolic acids in peanut root exudates promoted change in the microbial community of the rhizosphere soil and played an important role in soil diseases of the peanut [13]. These studies have shown that allelochemicals have significant effects on soil microorganisms, especially several specific fungi and bacteria that can cause disease or release nutrients. Allelopathic substances released by plant roots can affect the microbial community in the rhizosphere soil, and the soil microbial community can also affect the allelopathy between plants and soil to a certain extent [14,15]. Therefore, exploring the correlation between allelochemicals and soil microorganisms is an important part of exploring plant allelopathy.

Soil enzyme activity is a sensitive index for soil environmental change; it is often related to soil physical and chemical properties, reflects the direction and intensity of soil biochemical processes, and has an important impact on soil physical and chemical properties, fertility, and biological conditions. Therefore, it is often used as an important index for evaluating soil environmental quality [16,17]. The correlation between soil microorganisms and soil enzymes has received increasing attention from scholars [18]. Studies have shown that the environmental factors of soil, such as altitude, pH, organic matter, total nitrogen, alkali-hydrolyzable nitrogen, available potassium, and available phosphorus content, were related to the composition and diversity of the soil microbial communities [19], which also affect the soil enzyme activity [20]. Each soil enzyme and environmental factor has a different correlation to the rhizosphere soil microbial community of *S. chamaejasme*. Jin studied the bacterial community in the rhizosphere and root of *S. chamaejasme* on the Qinghai–Xizang Plateau and found that soil phosphorus, pH, latitude, altitude, and potassium were positively correlated with the bacterial community in the rhizosphere soil [21]. However, it is still unknown how soil enzyme activities, environmental factors, and allelochemicals interact to impact the rhizosphere soil microbial community.

*Stellera chamaejasme* L., a perennial herb of *Stellera*, is widely distributed in Gansu, Xinjiang, Ningxia, and other provinces and regions in China. It has become one of the iconic grassland degradation plants in China due to its predominant ecological adaptability and high competitiveness [21]. Studies suggest *S. chamaejasme* can release flavonoid allelochemicals to restrain the growth of other plants [22,23,24]. Pharmacological activity [25], chemical substances [26], ecology [27], and a few reports on rhizosphere soil microorganisms [21,28] are the bulk of the current research on *S. chamaejasme*. However, neither allelochemicals in the rhizosphere soil nor the soils physical and chemical properties’ interaction to impact soil microorganisms have been reported.

Allelochemicals released by plants during invasion may be an important factor in the competition of habitat expansion. Allelochemicals released by *S. chamaejasme* may be the main mechanism for its invasion of the process affecting the rhizosphere soil microbial community structure. However, the correlation between the microbial community structure and allelochemicals in the rhizosphere of *S. chamaejasme* is not clear. This study focused on how the soil enzyme activities, environmental factors, and allelochemicals interact to impact the soil microbial community in the rhizosphere of *S. chamaejasme* on the representative natural alpine steppe in the eastern foot of Qilian Mountain. Our purpose is to explore the correlation between soil enzymes, physical and chemical properties, and allelochemicals with soil microorganisms in different growth gradients. With the help of high-throughput sequencing technology and high-performance liquid chromatography (HPLC), we hope to explore the interactions among the microbial community, enzyme activity, environmental factors, and allelochemicals in the *S. chamaejasme* rhizosphere soil. We hope to reveal the interactive relationships that affect the *S. chamaejasme* rhizosphere microecosystem’s survival competition and ecological adaptation as well as provide a theoretical basis for scientific and effective ecological control strategies for poisonous weeds.

## 2. Materials and Methods

### 2.1. Sampling Location Information

The sample collection time was during the flowering stage of *S. chamaejasme* in July 2019. The sample plot was located in Tanyaogou Village, Xiulong Township, Tianzhu Tibetan Autonomous County, Wuwei City, Gansu Province (Table 1). The sample plots were selected according to the four growth coverages of *S. chamaejasme* at 0% (no invasion), 25.13% (primary invasion), 52.63% (moderate invasion), and 89.69% (severe invasion), and four replicates were set for each plot covered by growth coverage. The distance between each sample site was more than 100 m, and the distance between individual plants was more than 10 m. Fifteen plants that looked healthy and had similar characteristics were collected with aseptic shovel and gloves in each plot. Rhizosphere soil samples were collected by shaking the root in a sterile plastic bag [28]. The rhizosphere soils of each growth coverage were fully mixed and made into composite samples, which were returned to the laboratory at −4 ℃ for preservation and retention.

### 2.2. Soil Chemical Analysis

After shade drying, slight grinding, and impurity removal through a 0.4 mm sieve, the rhizosphere soil samples were sent to the Central Laboratory of Lanzhou Mineral Exploration Institute (http://www.gsyslky.com, accessed on 3 March 2022) for determination of soil characteristics, including pH value, organic matter, total nitrogen, alkali-hydrolyzable nitrogen, available potassium, and available phosphorus. Soil enzyme activity is measured by using a soil enzyme kit (Suzhou Comin Biotechnology Co., Ltd., Suzhou, Jiangsu, China) and an applicable UV-Vis spectrometer (UV-1750, Shimadzu Co., Ltd., Kyoto, Japan). All enzyme activities in the rhizosphere soil were determined by air-drying mass.

### 2.3. Quantification of Chemical Substances in Rhizosphere Soil

We shade dried and sieved the collected rhizosphere soil, and then removed plant roots, stones, and other impurities. We accurately weighed 100 g rhizosphere soil and extracted it repeatedly with 300 mL methanol 3 times with the aid of ultrasonic wave, each time for 30 min. We filtered the extract, steamed it dry it with a rotary evaporator, and collected the residue. The residue was dissolved in chromatographic methanol and passed through 0.22 μm filter membrane for quantitative analysis. HPLC was carried out with a Waters^®^ Breeze™ 2 System instrument and Breeze 2 Software Add-On System; the analytical column was a 250 mm × 4.6 mm, 5 μm particle size Symmetry C18 reversed-phase column. The solvent of acetonitrile for HPLC analysis was of HPLC gradient grade (Anhui Fulltime Co., Ltd., Hefei, China). Ultrapure water was obtained from Hangzhou Wahaha Co., Ltd. (China) and recorded the ultraviolet spectrum at 300 nm. The flow rate was 1 mL/min, the injection volume was 20 μL, and the column temperature was 30 °C. Mobile phase A was acetonitrile, and phase B was water (0.2% acetic acid). This gradient was followed: 0–6 min, 20–30% A; 6–8 min, 30–35% A; 8–15 min, 35–45% A; 15–18 min, 45–60% A; 18–20 min, 60–80% A; 20–25 min, 80% A.

The chemicals in the rhizosphere soil were determined by comparing the retention time and ultraviolet spectrum of the standard. The allelochemicals secreted by *S. chamaejasme* were detected by the internal standard method of HPLC. The peak with the same retention time and ultraviolet spectral characteristics was considered to be the same allelopathic substance. We compared the measured peak area of the sample and the peak area of the standard product to estimate the allelochemical content in the rhizosphere soil [23].

### 2.4. High-Throughput Sequencing of Soil Microorganisms

The total genomic DNA was extracted from the rhizosphere soil by MoBio kit. Then, 1% agarose gel electrophoresis was used to detect the purity and concentration of the DNA, and the diluted DNA group was used as a template. In terms of fungi, the ITS1 region of the fungus was amplified by PCR, and the primers were sequenced as ITS1F (5′CTTG GTCA TTTA GAGG AAGT AA-3′) [29] and ITS2 (5′GCTG CGTT CTTCA TCGA TGC-3′) [30]. The 16s rRNA V3-V4 region of bacteria was amplified by PCR, and the primers were 341F (5′-ACTC CTAC GGGA GCAG CAGC AG-3′) and 806R (5′-GGAC TACH VGGG TWTC TAAT-3′) [31]. Specific barcode sequence tags were added to distinguish different samples. The PCR reaction system 5×Fast Pfu Buffer was 4 μL, the dNTPs (2.5 mmol/L) was 2 μL, the positive and reverse primers (5 μmol/L) were 0.8 μL each, the fast Pfu polymerase was 0.4 μL, and the template DNA was 10 ng, supplemented to 20 μL with ddH2O. The PCR reaction conditions of fungi were as follows: 94 °C, 20 min, 55 °C, 55 °C, 72 °C, 30 cycles, 72 °C, 5 min. The bacterial PCR reaction conditions were as follows: 94 °C 3 min; 94 °C 30 s 55 °C 30 s 72 °C 30 s, 25 cycles. The PCR products were detected by 2% agarose gel electrophoresis, recovered by a GeneJET (Thermo Scientific, Shanghai, China) gel recovery kit, and Illumina high-throughput sequencing was carried out by Shanghai Shenggong Bioengineering Co., Ltd.

### 2.5. Data Processing and Analysis

The high-throughput sequencing data were removed by Cutadapt software, and the pairs of reads were spliced by PEAR software; each sample data was identified and distinguished by barcode tag sequences. Finally, the quality of each sample data was filtered by PRINSEQ software, and effective data were obtained. Using USEARCH to remove the non-amplified region sequence UCHIME to identify chimerism, and BLAST alignment on the representative sequence of the database, the outer sequence of the target region below the threshold value of 0.8 was removed, and then the operational taxonomic units (OTUs) were divided according to 97% similarity by USEARCH software. Fungi and bacteria were compared with SILVA (http://www.arb-silva.de/, accessed on 21 March 2022) [32] and RDP (http://rdp.cme.msu.edu/misc/resources.jsp, accessed on 23 March 2022) databases, respectively, to obtain the species classification information corresponding to each OTU [33].

Taking OTU as the object, a rarefaction analysis was completed by using a software called mothur; the sparsity curve was drawn by R, and the α diversity index of the microbial community was calculated. This included the Shannon index and Simpson index, which represent the diversity of community distribution. Chao1 estimator and ACE estimator, which represent the richness of soil microbial community and coverage index, indicated the sequencing depth. The representative sequence of OTUs was selected to annotate and classify the microorganisms in different samples, and R was used to map the statistical results of species taxonomy. All data were checked by SPSS 26 Software for Windows to test whether they met the normal distribution and then complete statistical analysis. The statistical data used in this study were processed by IBM SPSS Statistics 26. A Pearson correlation analysis method with the SigmaPlot 12.5 tool was used to study the relationship between rhizosphere soil microbial community index and allelopathic substances, soil enzyme activities, and physical and chemical parameters (Systat Software, Inc., San Jose, CA, USA). Based on the correlation similarity matrix, principal component analysis was carried out by R software to analyze rhizosphere soil microbial community. Redundancy analysis (RDA) in Canoco5.0 was used to explore the correlation among environmental factors, soil enzyme activities, and allelochemicals.

### 2.6. Nucleotide Sequence Accession Numbers

The representative bacterial sequences generated in this study were submitted to GenBank under the following accession numbers: SRR14339806, SRR14339807, SRR14339808, SRR14339809, SRR14339811, SRR14339812, SRR14339813, SRR14339814, SRR14339815, SRR14339816, SRR14339817, and SRR14339818. The accession numbers of the representative fungal sequences were: SRR14339799, SRR14339800, SRR14339801, SRR14339802, SRR14339803, SRR14339804, SRR14339805, SRR14339810, SRR14339819, SRR14339820, SRR14339821, and SRR14339822.

## 3. Results

### 3.1. Soil Physical and Chemical Properties and Enzyme Activity

The physical and chemical properties of the soil at the sampling sites were different under different coverages (Table 2). The range of soil pH in the rhizosphere of *S. chamaejasme* was 7.46–7.61 under four cover degrees, and the soil pH increased when the *S. chamaejasme* growth coverage increased. Compared to the plots without *S. chamaejasme*, the contents of organic matter, total nitrogen, alkali-hydrolyzable nitrogen, available potassium, and available phosphorus in the plots with *S. chamaejasme* were relatively higher. Among them, the sample plots with 25.13% coverage had the highest contents of total nitrogen, alkali-hydrolyzable nitrogen, available potassium, and available phosphorus, followed by the sample plots with 89% coverage. 

The trends of enzyme activities in seven kinds of rhizosphere soil were different in the four plots. The activities of peroxidase (POD) and dehydrogenases (DHA) have a similar trend in soil; that is, they have the same changing trend as the coverage changes. In the sample plots with *S. chamaejasme* growth, the activities of polyphenol oxidase (PPO), POD, and DHA in the rhizosphere soil were higher than the soil without *S. chamaejasme*. However, urease (UE), sucrose (SC), acid phosphatase (ACP), and alkaline phosphatase (AKP) were lower than those without *S. chamaejasme*.

### 3.2. Quantitative Analysis of Allelochemicals in Rhizosphere Soil

Five flavonoids allelochemicals in rhizosphere soil were detected by HPLC; they were chamaechromone, mesoneochamaejasmin A, neochamaejasmin B, Dihydrodaphnodorin B, and 7-methoxyneochamaejasmine (Table 2). The results showed that the content of five allelochemicals was relatively high in the plot with 52.63% growth coverage, followed by the plot with 89.69% growth coverage, and relatively low content was found in the plot with 25.13% growth coverage. The concentrations of neochamaejasmin B and 7-methoxyneochamaejasmine A were higher than the other three allelochemicals. As can be seen in the picture, with the increase in the growing coverage of *S. chamaejasme*, the quality of allelochemicals secreted into the rhizosphere soil showed an increasing trend. When the growth coverage was more than 52.16%, the release of allelochemicals decreased (Table 2, Figure 1).

### 3.3. Analysis of High-Throughput Sequencing Data

We divided the samples with the same coverage into three repeated controls for high-throughput sequencing analysis and then summarized the data. Four samples from the rhizosphere soil of *S. chamaejasme* with different coverage were clustered with more than 97% similarity to obtain 3788 fungal OTU and 15,496 bacterial OTU. Good coverage of fungi or bacteria included more than 99% or 93%, respectively. Additionally, the sample rarefaction curve shows (Figure 2) that with the increase in the number of sequencing samples, the four samples’ OTU rarefaction curves tend to be smooth. This showed that the amount of data sequenced in this experiment was gradually reasonable and comprehensively reflected the microbial community composition. The increase in the amount of data contributes less to discovering new OTU numbers.

As the species composition of the *S. chamaejasme* rhizosphere soil fungi at the phylum level under different growth coverage shows, 12 phyla of fungi were obtained from four rhizosphere soil samples with different growth coverages (Figure 3a). Basidiomycetes were the main dominant fungi in the code “No” sample, accounting for 65.31% of the total fungi sequence in the samples. This was then followed by Ascomycetes, accounting for 29.08%. Among the other plot types with *S. chamaejasme* growing, Ascomycetes were the main dominant phyla, accounting for 52.30%, 49.49%, and 50.88% of the total sequence, followed by basidiomycetes, accounting for 39.35%, 41.79%, and 39.16%. Additionally, 32 phyla of bacteria were obtained; the differences in the bacterial composition in different coverage plots were similar (Figure 3b). The main dominant bacterial phylum was Proteobacteria, accounting for 35.11%, 43.34%, 42.34%, and 40.90% of the total sequence at the phylum level. Additionally, the dominant bacteria were Acidobacteria, Actinobacteria, Planctomycetes, and Verrucomicrobia.

At the genus level, the composition of the microbial community in rhizosphere soil with different coverages of *S. chamaejasme* is shown in the bar plot (Figure 4). We selected the top 30 fungi and bacteria with the highest abundance and made statistical maps. *Inocybe* was the main genus of fungi in the rhizosphere soil of the 0% plot, accounting for 27.61%. Additionally, *Cortinarius*, *Archaeorhizomyces,* and *Sebacina* accounted for 10.21%, 6.21%, and 5.47%, respectively. The composition of the fungal genus was similar in the samples growing *S. chamaejasme*, and the main dominant fungal genus was *Archaeorhizomyces*, followed by *Sebacina*, *Inocybe,* and *Mortierella* (Figure 4a). The composition structure of the bacterial community was similar in the rhizosphere soil of different coverage plots with *S. chamaejasme*. However, among the samples with 0% coverage, the abundance of Gp4 was the highest, accounting for 21.56% of the top 30 species. Additionally, this was followed by Gp6 (14.30%), *Spartobacteria genera incertae sedis* (9.63%), and *Sphingomonas* (9.19%). Among the other three samples with *S. chamaejasme*, the order of abundance from high to low was *Sphingomonas* (17.14%, 15.46%, and 14.76%, according to coverage from a high quantity to low quantity), *Gp6* (14.80%, 14.7%, and 12.51%), Gp4 (10.43%, 11.64%, and 10.77%), and *Spartobacteria genera incertae sedis* (6.83%, 6.82%, and 9.98%) (Figure 4b).

Alpha diversity analysis of microorganisms in the rhizosphere soil of *S. chamaejasme* is shown in Table 3. Chao1 index and ACE index increased with the increase in coverage, while the Simpson index decreased with the increase in coverage. Rhizosphere soil bacteria have the same changing trend as fungi. To synthesize the above, as the *S. chamaejasme* growth coverage increased, the richness and diversity of the fungi and bacteria in the sample plot increased. The Shannon diversity index also showed that the diversity of the bacterial community in the rhizosphere soil samples was higher than that of the fungi. With the increase in the growth coverage of *S.chamaejasme*, the Shannon diversity index showed an increasing trend (Figure 5).

### 3.4. Relationship between Soil Environmental Factors, Soil Enzyme Activity, and Allelopathic Substances and Bacterial Community

Pearson correlation analysis revealed that Basidiomycetes were negatively correlated with AP and XB and positively correlated with SC. Ascomycetes were positively correlated with AP and negatively correlated with SC. Conversely, Mortierellomycota was positively correlated with AK (Table 4). Bacterial Pearson correlation analysis shows that Proteobacteria was negatively correlated with UE and ALP and positively correlated with SYT and XB. Acidobacteria was negatively correlated with soil pH and AP and positively correlated with SH and ALP. Actinobacteria was positively correlated with soil pH, SYT, and XB and negatively correlated with SH and UE. Planctomycetes were positively correlated with UE and ACP and negatively correlated with allelochemicals SYT, XB, and QB (Table 5). It seemed that enzyme activity, environmental factors, and allelochemicals in rhizosphere soil have more effects on the bacterial community.

Redundancy analysis (RDA) was used to explain the microbial community (response variables) using soil enzyme activities, allelochemicals, and environmental factors (explanatory variables) at the different study sites. A Monte Carlo test of fungi and bacteria showed *p* < 0.05, indicating a linear relationship between the environmental factors and rhizosphere soil microorganisms. The fungi RDA analysis showed that the RDA1 was 99.3%, and the RDA2 was 0.4%, which can better reflect the relationship between enzyme activity, environmental factors, and the allelopathic and soil fungal community (Figure 6a). Basidiomycota was related to SH, At, ALP, UE, and SC and had a certain correlation with SOM. Mortierellomycota and Glomeromycota were related to soil DHA and POD. Glomeromycota had a certain correlation with PPO and ST. Five allelochemicals were positively correlated with Ascomycota and negatively correlated with Basidiomycota. The bacteria RDA analysis showed that the RDA1 was 92.0%, and the RDA2 was 7.2% (Figure 6b). Acidobacteria and Planctomycetes were highly correlated with soil SH, SC, ALP, UE, ACP, and At and were correlated with SOM. Proteobacteria and Actinobacteria were correlated with soil pH, AP, PPO, POD, and DHA, as well as all allelochemicals. However, there was little correlation between Proteobacteria, Actinobacteria, and the contents of POD, AN, TN, and AK in soil.

## 4. Discussion

An invasion of plants will release a variety of compounds, and some of the allelochemicals released into soil will impact other plants around them. Some studies have shown that *S. chamaejasme* releases different amounts of flavonoids into the soil through its roots [23]. The root is one of the main ways that plants release flavonoids to the outside world [34], and flavonoids can adapt plants to environmental stresses, including biological stress and abiotic stress [35]. Several studies have shown that flavonoids secreted by plants into rhizosphere soil can inhibit the growth of surrounding plants [36]. Yan confirmed that the flavonoid allelochemicals released by *S. chamaejasme* have an obvious inhibitory effect on Arabidopsis thaliana seedlings [24]. This paper is the first study on the rhizosphere allelochemicals and microbial diversity of *S. chamaejasme* with different coverages. When plants secrete flavonoid allelochemicals to inhibit the growth of other weeds, autotoxicity easily occurs with the extension of the planting time [37].

Using the high-throughput sequencing technique, we found that the similarity of microbial composition in the growing area of *S. Stellera* was higher, and, the higher the community coverage was, the closer the similarity was. With the increase in the growth coverage, the abundance and diversity of microorganisms in the rhizosphere soil showed an increasing trend (Table 3, Figure 5). In addition, Basidiomycetes and Ascomycetes were the main fungi in the soil, while Proteobacteria and Acidobacteria were the main bacteria. This was similar to the investigation and study of Minxian County and Cuiying Mountain in Gansu Province by Jin and others [28,38]. By comparison, Basidiomycetes was the dominant phylum in the rhizosphere soil at a 0% *S. chamaejasme* cover. The invasion of *S. chamaejasme* might result in the dominance of the Ascomycetes in the soil, and the growth of *S. chamaejasme* increases the enrichment of Proteobacteria and reduces the composition of Acidobacteria in the rhizosphere soil. It seemed that the invasion of *S. chamaejasme* destroyed the original microbial composition and made it more conducive to its own expansion. Basidiomycetes and Ascomycetes were important decomposers in soil, and under the action of enzymes, they can decompose complex organic compounds, including polycyclic aromatic hydrocarbons [39,40]. Proteobacteria can significantly promote the cycle of nitrogen, phosphorus, sulfur, and organic matter in soil [41], and some studies have shown that acid bacilli have a certain ability to decompose cellulose [42]. These were the main components of the microorganisms in the rhizosphere of *S. chamaejasme* and played an important role in their growth, development, and invasion.

Studies have shown that allelochemicals can change the growth of neighboring plants and soil microbial communities [43]. Currently, research on rhizosphere fungi and allelochemicals of *S. chamaejasme* has not been reported. The influence of plant growth coverage on the secretion of allelochemicals is also unknown. With the increase in the growth coverage of *S. chamaejasme*, the quality of allelochemicals released into the rhizosphere soil showed an increasing trend. When the growth coverage was more than 52%, the release of allelochemicals decreased (Table 2, Figure 1). Therefore, the increase in growth coverage weakens the secretion of allelochemicals once it reaches a threshold. The results of Pearson correlation analysis in this study showed that bacteria were more greatly affected by allelochemicals. RDA analysis showed that Ascomycota and Glomeromycota were positively correlated with neochamaejasmin B, chamaechromone, and dihydrodaphnetin B, indicating that these three allelochemicals had important effects on Ascomycota and Glomeromycota. In addition, neochamaejasmin B and chamaechromone were highly correlated with Proteobacteria and Actinobacteria of bacteria. This may be due to the fact that neochamaejasmin B, chamaechromone, and dihydrodaphnetin B promote the competitiveness of microorganisms in the rhizosphere of *S. chamaejasme*. Studies have shown that the invasion of exotic plants may change the structure and function of some microorganisms [44]. Ni found that Centaurea diffusa can also release 8-hydroxyquinoline through its root system, which can change the microbial community in susceptible soil [45]. It can also improve the competitiveness of invasive plants. Therefore, *S. chamaejasme* secretes allelochemicals to affect soil microorganisms and create a rhizosphere microbial community suitable for its own growth, so as to enhance its invasion competitiveness. In addition, the metabolism of flavonoids by the soil microbial community may change the relative abundance of some native microbial species, the activity of the microbial population, and the availability of pollutants in the soil [46]. Vivanco et al. found that allelopathic substances were decomposed and transformed by soil microorganisms after entering the soil, resulting in allelopathic effects on the surrounding plants [47]. Therefore, the interaction between the allelochemicals and microorganisms in rhizosphere soil may play an important role in the invasion of *S. chamaejasme*.

Soil enzyme activity is one important indicators that reflects changes in the natural environment [48]. The results of Pearson correlation analysis showed that there was little correlation between soil microorganisms and seven kinds of soil enzyme activities. RDA analysis showed a significant positive correlation between DHA and soil allelochemicals, which affected some soil microorganisms. The results show that dehydrogenase can characterize the activity of soil microorganisms [49]. Some studies have also shown that root exudates change the microbial community structure and soil enzyme activity in rhizosphere soil [50]. Therefore, the soil microbial community structure of *S. chamaejasme* is not only related to allelochemicals but also affects the activity of soil enzymes. In this study, we found that the increase in the growth coverage of *S. chamaejasme* would increase the activity of some soil enzymes and weaken the activity of other soil enzymes. When we compared the changes in enzyme activity in the soil with different growth coverages, we found that the enzyme activity has a greater change in soil without the growth of *S. chamaejasme*. Secondly, when the growth coverage was about 25%, the change in enzyme activity was the largest. When the coverage increases, the changes in soil enzyme activity tend to be gradual. Therefore, the invasion of *S. chamaejasme* greatly affects soil enzyme activity in the initial stage.

Soil environmental factors play an important role in determining the composition of soil microbial communities [51]. Pearson analysis showed that the correlation between soil physical and chemical properties and microbial community was low. Principal component analysis showed little correlation between these and soil microorganisms. In RDA analysis, we found a great correlation between environmental factors and allelochemicals, and this was consistent with the trend of allelopathic substances affecting the microbial community. Mulderij found that the allelopathy of submerged plants was obvious under nutrient stress [52]. Yu found that the soil NH_4_^+^, NO_3_^−^, and available P and K in the heavily invasive soil of *Eupatorium adenophorum* were significantly higher than those in the slightly invasive soil, and there were significant differences in the characteristics of the soil bacterial community [53]. In the invaded soil of *S. chamaejasme*, the physical and chemical properties gradually increased with the growth coverage, and it had a higher content of nutrient elements. In addition, similar to the soil enzyme activity results, we found that when the growth coverage was 25%, the rhizosphere soil had a higher content of TN, AN, AK, and AP. Therefore, the initial invasion stage of *S. chamaejasme* affects the physical and chemical properties of soil and the content of various nutrient elements.

## 5. Conclusions

In the soil invaded by *S. chamaejasme*, the physical and chemical properties gradually increased with the growth coverage, and the soil had a higher content of nutrient elements. In the initial invasion stage, *S. chamaejasme* obviously affected the nutrient element content and enzyme activity of the soil. The invasion of *S. chamaejasme* changed the microbial community structure of the original soil. As the growth coverage gradually increased, the microbial community structure became more similar. When the microbial community changed, it increased the soil nutrient elements around the rhizosphere of *S. chamaejasme* and improved the soil physical and chemical properties. It also changed the soil enzyme activity. The effects of the initial invasion stage were greater than in the other invasion stages. The release of allelopathic substances increases with the increase in growth coverage. When the growth coverage exceeds 52%, the secretion of allelopathic substances into the soil is inhibited.

## Figures and Tables

**Figure 1 microorganisms-10-00158-f001:**
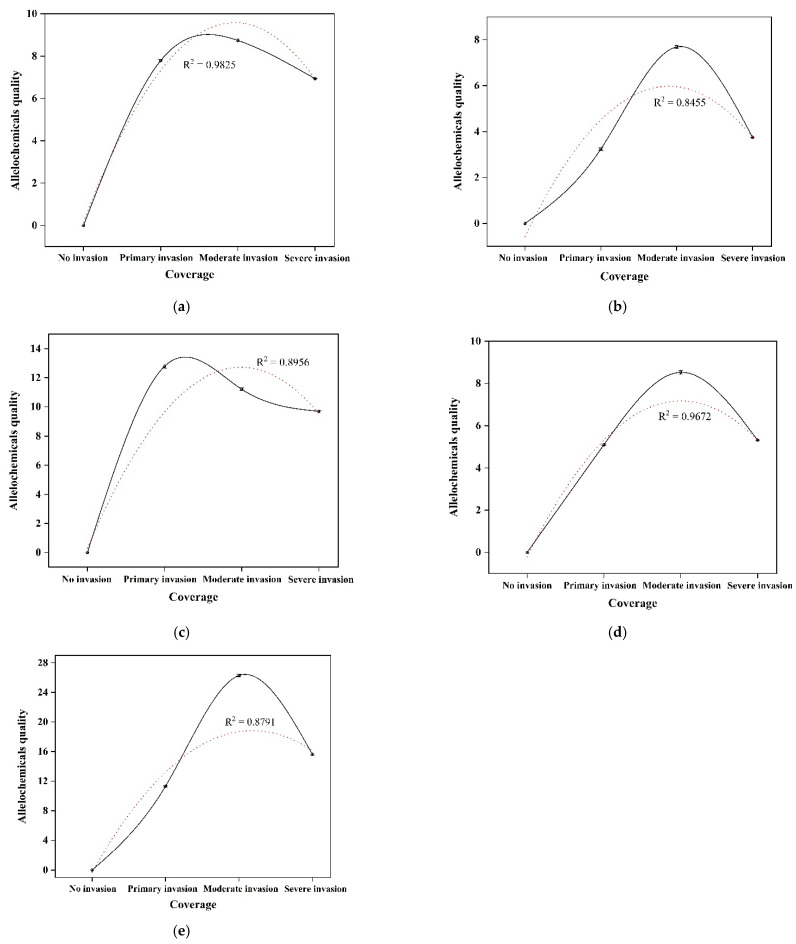
The relationship between growth coverage and quantity of allelochemicals: (**a**) chamaechromone; (**b**) mesoneochamaejasmin A; (**c**) neochamaejasmin B; (**d**) dihydrodaphnodorin B; and (**e**) 7-methoxyneochamaejasmine A.

**Figure 2 microorganisms-10-00158-f002:**
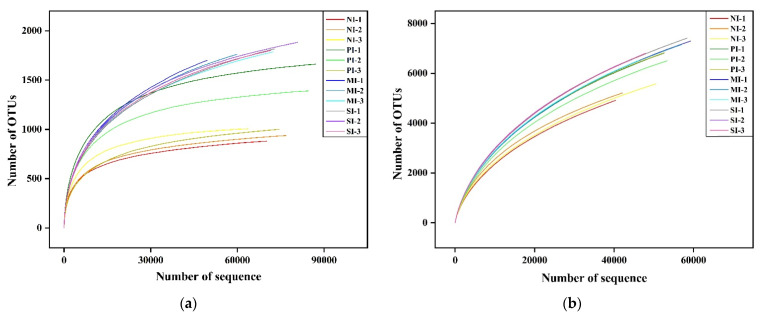
Fungi (**a**) and bacteria (**b**) rarefaction curves. NI, no invasion; PI, primary invasion; MI, moderate invasion; SI, severe invasion. OTUs, operational taxonomic units.

**Figure 3 microorganisms-10-00158-f003:**
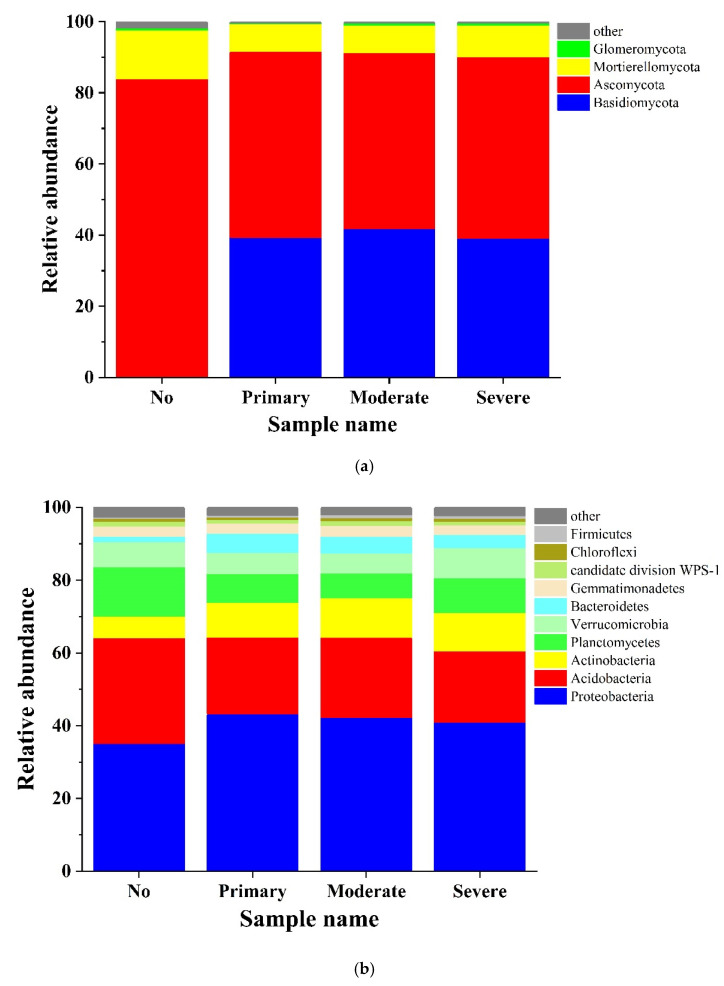
Phylum composition of fungi (**a**) and bacteria (**b**). No, no invasion; primary, primary invasion; moderate, moderate invasion; severe, severe invasion.

**Figure 4 microorganisms-10-00158-f004:**
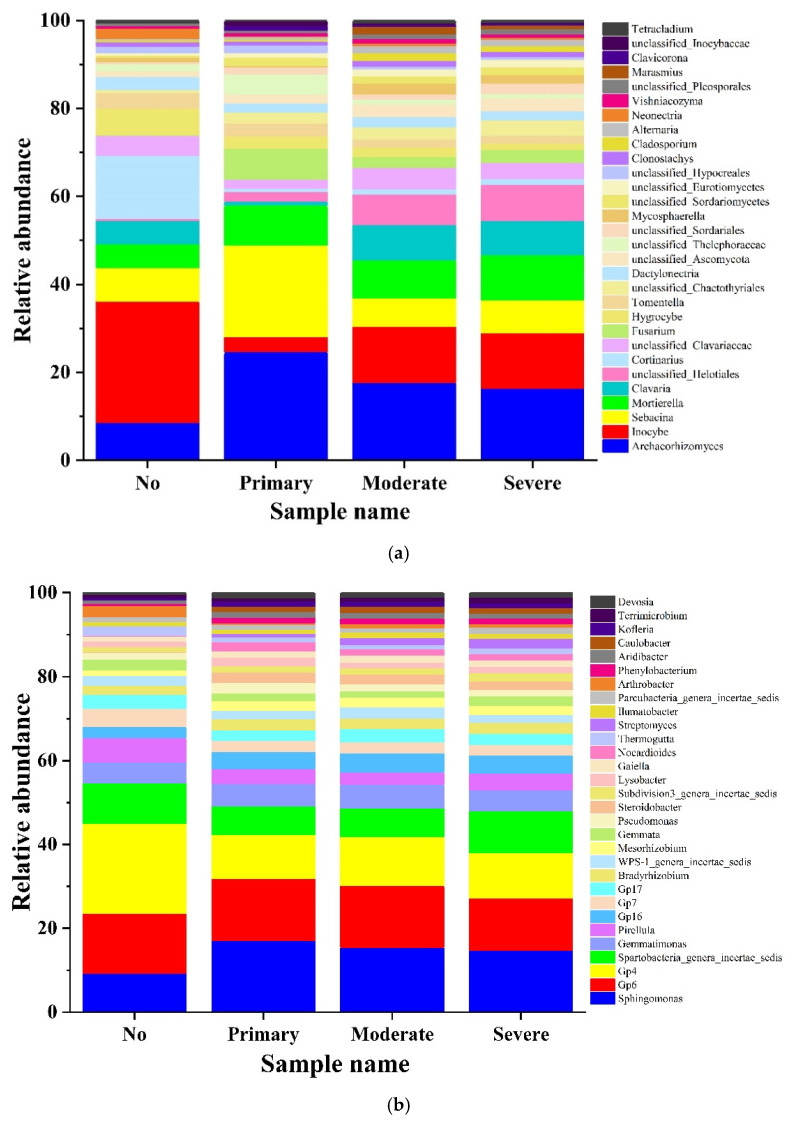
Genus composition of fungi (**a**) and bacteria (**b**).

**Figure 5 microorganisms-10-00158-f005:**
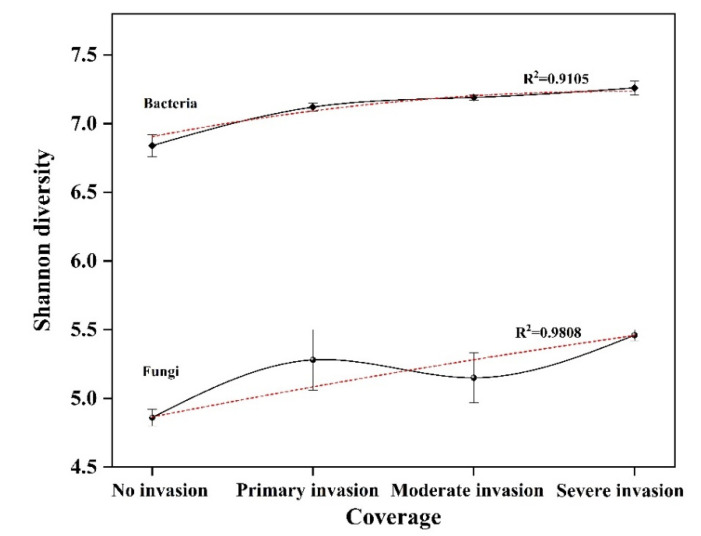
Shannon diversity of fungi and bacteria varies with coverage.

**Figure 6 microorganisms-10-00158-f006:**
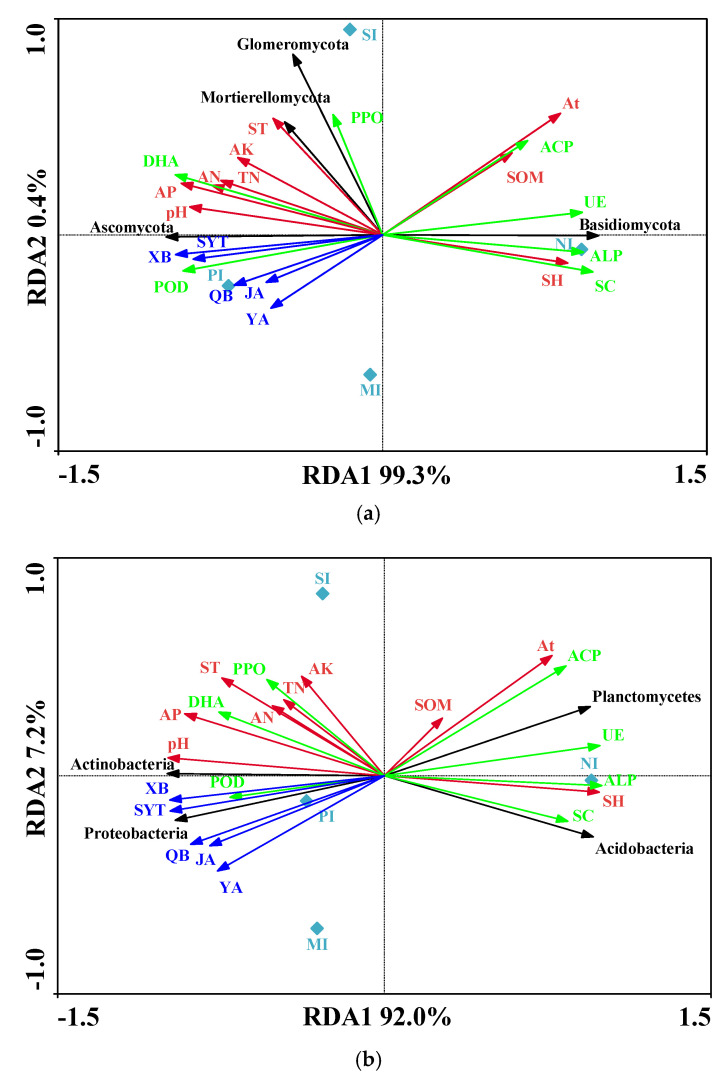
Redundancy analysis (RDA) of fungi (**a**) and bacteria (**b**). The red arrow and the green arrow represent the relative position of soil physical and chemical properties and enzyme activity on the horizontal plane; the blue arrow represents the relative position of soil allelochemicals on the horizontal plane. The black arrow represents the species distribution at the phylum level, and the longer the arrow, the greater the impact of the species in the sample. Where the angle between the arrow and the sort axis is different, the correlation strength is also different. The smaller the angle, the greater the correlation, and, the longer the length of the arrow, the greater the effect of the environmental factor. Note: pH, hydrogen ion concentration; At, altitude; ST, soil temperature; SH, soil humidity; SOM, soil organic matter; TN, total nitrogen; AN, alkali-hydrolyzable nitrogen; AK, available potassium; AP, available phosphorous; PPO, polyphenol oxidase; POD, peroxidase; UE, urease; DHA, dehydrogenases; SC, sucrase; ALP, alkaline phosphatase; ACP, acid phosphatase; SYT, chamaechromone; YA, mesoneochamaejasmin A; XB, neochamaejasmin B; QB, dihydrodaphnodorin B; JA, 7-methoxyneochamaejasmine A; NI, no invasion; PI, primary invasion; MI, moderate invasion; SI, severe invasion.

**Table 1 microorganisms-10-00158-t001:** Sample plot information table.

Sample Number	Altitude (m)	Northern Latitude	Eastern Longitude	Population Coverage	Coverage Gradient (%)
1	2960	37°7′37″	102°50′3″	0.00	0
2	2970	37°7′36″	102°50′3″	0.00	
3	2970	37°7′33″	102°49′38″	0.00	
4	2970	37°7′33″	102°50′0″	0.00	
5	2960	37°7′36″	102°50′4″	25.50	25.13 ± 1.26
6	2890	37°7′33″	102°49′56″	26.75	
7	2930	37°7′37″	102°50′17″	25.00	
8	2940	37°7′36″	102°50′17″	23.25	
9	2940	37°7′34″	102°50′1″	50.50	52.63 ± 2.76
10	2920	37°7′42″	102°50′14″	57.25	
11	2930	37°7′48″	102°50′16″	50.50	
12	2950	37°7′42″	102°50′5″	52.25	
13	2960	37°7′35″	102°50′2″	85.00	89.69 ± 4.81
14	2970	37°7′35″	102°49′59″	85.75	
15	2950	37°7′43″	102°50′12″	97.00	
16	2950	37°7′45″	102°50′16″	91.00	

**Table 2 microorganisms-10-00158-t002:** Statistical table of physical and chemical properties, enzyme activities, and allelochemicals in rhizosphere soil.

Name	No Invasion	Primary Invasion	Moderate Invasion	Severe Invasion
pH	7.46 ± 0.01 b	7.60 ± 0.01 a	7.60 ± 0.02 a	7.61 ± 0.01 a
At (m)	2968 ± 2.50 a	2930 ± 14.72 b	2935 ± 6.45 b	2958 ± 4.79 a
ST (°C)	4.80 ± 0.15 a	4.93 ± 0.09 a	5.03 ± 0.09 a	5.17 ± 0.09 a
SH (%)	39.39 ± 2.56 a	30.48 ± 0.34 b	29.63 ± 0.18 b	29.18 ± 0.37 b
SOM (g/kg)	131.41 ± 0.94 b	113.83 ± 0.29 c	132.11 ± 0.52 b	135.45 ± 0.85 a
TN (g/kg)	6.83 ± 0.04 c	8.24 ± 0.01 a	6.66 ± 0.01 d	7.45 ± 0.01 b
AN (mg/kg)	586.23 ± 0.96 c	690.35 ± 2.56 a	582.06 ± 1.91 c	634.12 ± 2.74 b
AK (mg/kg)	255.4 ± 0.68 c	368.1 ± 1.36 a	223.42 ± 1.99 d	310.28 ± 1.88 b
AP (mg/kg)	38.4 ± 0.21 d	58.94 ± 0.17 a	49.97 ± 0.41 c	56.91 ± 0.23 b
PPO (mg/d/g)	13.53 ± 0.12 c	13.74 ± 0.12 c	16.03 ± 0.10 b	17.61 ± 0.05 a
POD (mg/d/g)	30.44 ± 0.17 c	45.59 ± 0.26 b	35.79 ± 0.06 a	35.59 ± 0.46 b
UE (μg/d/g)	860.93 ± 1.41 a	756.62 ± 0.29 c	758.08 ± 0.41 c	776.10 ± 0.48 b
DHA (μg/d/g)	6.09 ± 0.08 d	23.03 ± 0.16 a	10.85 ± 0.13 c	17.33 ± 0.15 b
SC (mg/d/g)	62.81 ± 0.17 a	59.79 ± 0.16 d	61.56 ± 0.13 b	60.74 ± 0.31 c
AKP (umol/d/g)	8.11 ± 0.13 a	4.25 ± 0.06 b	4.55 ± 0.06 b	4.41 ± 0.14 b
ACP (umol/d/g)	15.63 ± 0.10 a	13.99 ± 0.10 b	12.99 ± 0.20 c	14.45 ± 0.15 b
SYT (mg/kg)	--	7.79 ± 0.06 b	8.74 ± 0.08 a	6.93 ± 0.02 c
YA (mg/kg)	--	3.24 ± 0.06 c	7.69 ± 0.07 a	3.75 ± 0.01 b
XB (mg/kg)	--	12.78 ± 0.13 a	11.21 ± 0.09 b	9.69 ± 0.05 c
QB (mg/kg)	--	5.10 ± 0.04 b	8.52 ± 0.10 a	5.32 ± 0.01 c
JA (mg/kg)	--	11.31 ± 0.10 c	26.28 ± 0.20 a	15.62 ± 0.09 b

Note: pH, hydrogen ion concentration; At, altitude; ST, soil temperature; SH, soil humidity; SOM, soil organic matter; TN, total nitrogen; AN, alkali-hydrolyzable nitrogen; AK, available potassium; AP, available phosphorous; PPO, polyphenol oxidase; POD, peroxidase; UE, urease; DHA, dehydrogenases; SC, sucrase; ALP, alkaline phosphatase; ACP, acid phosphatase; SYT, chamaechromone; YA, mesoneochamaejasmin A; XB, neochamaejasmin B; QB, dihydrodaphnodorin B; JA, 7-methoxyneochamaejasmine A. a, b, c, d, significant difference.

**Table 3 microorganisms-10-00158-t003:** Alpha diversity of rhizosphere soil microorganisms.

Sample Name		Effective Tags	OTU	Shannon Index	Simpson Index	Chao1 Index	ACE Index	Coverage (%)
Fungi	F-0	70,240 ± 3823 b	938 ± 40 b	4.86 ± 0.06 b	0.026 ± 0.003 a	1039.00 ± 38.65 b	1028.29 ± 34.01 b	99.80
	F-1	81,977 ± 3836 a	1387 ± 201 a	5.28 ± 0.22 a	0.013 ± 0.003 a	1553.82 ± 208.18 a	1525.87 ± 203.69 a	99.08
	F-2	89,715 ± 13232 a	1429 ± 57 a	5.15 ± 0.18 a	0.017 ± 0.007 a	1678.31 ± 33.62 a	1665.56 ± 53.87 a	99.10
	F-3	77,406 ± 8489 a	1634 ± 92 a	5.46 ± 0.04 a	0.010 ± 0.000 b	1841.96 ± 90.65 a	1844.77 ± 97.90 a	99.30
Bacteria	B-0	44,335 ± 3130 a	5235 ± 191 b	6.84 ± 0.08 b	0.005 ± 0.000 b	8413.62 ± 309.63 b	10,702.66 ± 442.60 c	94.60
	B-1	52,847 ± 271 a	6713 ± 181 a	7.12 ± 0.03 a	0.004 ± 0.000 a	10,251.66 ± 116.19 a	12,729.96 ± 125.85 b	94.40
	B-2	54,133 ± 4029 a	6989 ± 414 a	7.19 ± 0.02 a	0.004 ± 0.000 a	10,912.19 ± 146.59 a	14,014.21 ± 220.17 a	94.10
	B-3	50,430 ± 6233 a	6940 ± 423 a	7.26 ± 0.05 a	0.003 ± 0.000 a	10,914.76 ± 254.76 a	13,567.63 ± 224.51 a	93.80

a, b, c, d, significant difference.

**Table 4 microorganisms-10-00158-t004:** Pearson correlation analysis of fungi community.

	Basidiomycota		Ascomycota		Mortierellomycota		Glomeromycota		Total Fungi		Diversity	
	CC	*p*	CC	*p*	CC	*p*	CC	*p*	CC	*p*	CC	*p*
pH	−0.906	0.094	0.875	0.125	0.160	0.840	0.633	0.367	0.760	0.240	0.939	0.061
At	0.821	0.179	−0.827	0.173	−0.021	0.979	0.100	0.900	−0.414	0.586	−0.602	0.398
ST	−0.533	0.467	0.477	0.523	0.027	0.973	0.903	0.097	0.819	0.181	0.864	0.136
SH	0.873	0.127	−0.838	0.162	−0.097	0.903	−0.642	0.358	−0.806	0.194	−0.962	0.038
SOM	0.573	0.427	−0.627	0.373	−0.583	0.417	0.324	0.676	0.397	0.603	0.106	0.894
TN	−0.729	0.271	0.764	0.236	0.915	0.085	0.298	0.702	−0.256	0.744	0.097	0.903
AN	−0.771	0.229	0.803	0.197	0.890	0.110	0.312	0.688	−0.200	0.800	0.155	0.845
AK	−0.651	0.349	0.686	0.314	0.961	0.039	0.342	0.658	−0.321	0.679	0.024	0.976
AP	−0.962	0.038	0.952	0.048	0.538	0.462	0.650	0.350	0.445	0.555	0.727	0.273
PPO	−0.258	0.742	0.194	0.806	−0.183	0.817	0.847	0.153	0.821	0.179	0.761	0.239
POD	−0.909	0.091	0.935	0.065	0.585	0.415	0.116	0.884	0.058	0.942	0.385	0.615
UE	0.934	0.066	−0.912	0.088	−0.099	0.901	−0.427	0.573	−0.715	0.285	−0.902	0.098
DHA	−0.923	0.077	0.936	0.064	0.747	0.253	0.485	0.515	0.128	0.872	0.466	0.534
SC	0.968	0.032	−0.975	0.025	−0.644	0.356	−0.489	0.511	−0.255	0.745	−0.577	0.423
ALP	0.941	0.059	−0.915	0.085	−0.202	0.798	−0.581	0.419	−0.710	0.290	−0.911	0.089
ACP	0.687	0.313	−0.657	0.343	0.355	0.645	−0.127	0.873	−0.822	0.178	−0.872	0.128
SYT	−0.89	0.110	0.861	0.139	0.001	0.999	0.437	0.563	0.787	0.213	0.940	0.060
YA	−0.539	0.461	0.495	0.505	−0.510	0.490	0.212	0.788	0.930	0.070	0.903	0.097

**Table 5 microorganisms-10-00158-t005:** Pearson correlation analysis of bacteria community.

	Proteobacteria		Acidobacteria		Actinobacteria		Planctomycetes		Total Bacteria		Diversity	
	CC	*p*	CC	*p*	CC	*p*	CC	*p*	CC	*p*	CC	*p*
pH	0.924	0.076	−0.979	0.021	0.997	0.003	−0.921	0.079	0.850	0.150	0.968	0.032
At	−0.914	0.086	0.586	0.414	−0.736	0.264	0.879	0.121	−0.215	0.785	−0.517	0.483
ST	0.530	0.470	−0.844	0.156	0.788	0.212	−0.602	0.398	0.983	0.017	0.930	0.070
SH	−0.905	0.095	0.970	0.030	−0.996	0.004	0.922	0.078	−0.864	0.136	−0.980	0.020
SOM	−0.488	0.512	0.179	0.821	−0.191	0.809	0.271	0.729	0.230	0.770	0.064	0.936
TN	0.526	0.474	−0.537	0.463	0.400	0.600	−0.272	0.728	0.280	0.720	0.278	0.722
AN	0.579	0.421	−0.581	0.419	0.455	0.545	−0.332	0.668	0.315	0.685	0.328	0.672
AK	0.424	0.576	−0.489	0.511	0.321	0.679	−0.162	0.838	0.276	0.724	0.228	0.772
AP	0.874	0.126	−0.959	0.041	−0.761	0.239	−0.761	0.239	0.781	0.219	0.841	0.159
PPO	0.289	0.711	−0.643	0.357	0.598	0.402	−0.423	0.577	0.890	0.110	0.798	0.202
POD	0.834	0.166	−0.652	0.348	0.651	0.349	−0.654	0.346	0.280	0.720	0.449	0.551
UE	−0.985	0.015	0.913	0.087	−0.982	0.018	0.979	0.021	−0.697	0.303	−0.889	0.111
DHA	0.780	0.220	−0.809	0.191	0.716	0.284	−0.589	0.411	0.563	0.437	0.612	0.388
SC	−0.860	0.140	0.866	0.134	−0.804	0.196	0.699	0.301	−0.612	0.388	−0.696	0.304
ALP	−0.953	0.047	0.972	0.028	−0.995	0.005	0.931	0.069	−0.805	0.195	−0.941	0.060
ACP	−0.861	0.139	0.663	0.337	−0.843	0.157	0.963	0.037	−0.468	0.532	−0.739	0.261
SYT	0.964	0.036	−0.902	0.098	0.985	0.015	−0.988	0.012	0.717	0.283	0.910	0.090
YA	0.736	0.264	−0.618	0.382	0.796	0.204	−0.895	0.105	0.530	0.470	0.759	0.241
XB	0.993	0.007	−0.916	0.084	0.971	0.029	−0.959	0.041	0.676	0.324	0.864	0.136
QB	0.856	0.144	−0.770	0.230	0.910	0.090	−0.964	0.036	0.646	0.354	0.861	0.139
JA	0.739	0.261	−0.684	0.316	0.835	0.165	−0.895	0.105	0.630	0.370	0.826	0.174

## Data Availability

All relevant data are contained within this article.

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
