# Peer review of "Effects of Allelochemicals, Soil Enzyme Activities, and Environmental Factors on Rhizosphere Soil Microbial Community of Stellera chamaejasme L. along a Growth-Coverage Gradient"

_microorganisms, 2022, doi:10.3390/microorganisms10010158_

Round 1
Reviewer 1 Report
The aims of manuscript are not easy to understand. Authors cannot reveal the internal mechanism that the effect of S. chamaejasme rhizosphere microecosystem on its survival competition and ecological adaptation based on these data. They can only describe some relations. Experiment was not designed in the best way, basically because soil was sampled in different localities (although not very distant) however soil microbiome is heavy spatially affected. The manuscript is very poorly written. Sometimes, hard to understand sentences can be found. Latin names are not in italics through whole manuscript.
Sample coding using numbers 0,1,2,3 is not well suited for reading graphs and other results, Authors should use wordy labels i.e. (no invasion, primary invasion, moderate invasion, severe invasion) or percentual coverage in graphs and tables.
L146-150 these are probably instructions from manual. Please rewrite it to the form suitable for publication.
L157 was any gradient used?
L159-165 HPLC without MS is not very accurate for determination of allelochemicals and many interferencies may be observed in soil extract and results may be haevily affected.
authors should not design novel names for primers they used ITS2 is the true primer name for GCTGCGTTCTTCATCGATGC. Also, why authors cite Usyk et al 2017 istead White et al 1990 for ITS2 and Gardes & Bruns, 1993 for ITS1f?
Table2 information about used statistics is missing. Sample coding should be corrected to give more sense for first insight
L252 how can be QUALITY gradually increased? I suppose it should be quantity... What is R value on graphs in figure 1? information about statistical interpretation is missing and reasoning why non-linear fit was chosen should be explained. Moreover 0,1,2,3 cannot be used for such analysis -- it is all wrong in my opinion and it must be redone using standard method after consultation with skilled statistician.
L261 horizontal coverage means Good`s coverage?
figure 2: Files in genbank under given accession numbers contains much lower number of sequences than presented in the figure. Where is the truth?
Table 3 again information about statistical test is missing. There is only statement about normal distribution testing in material and methods.
figure 5 statistics is used wrong X axis is wrong interpreted (categories instead percentage of coverage) the same fail is occurring along whole manuscript
Figure 6 the points for samples are missing
The manuscript needs significant improvement before publication
Author Response
Dear editor,
Thank you for your helpful comments and suggestions on our manuscript. We have revised the manuscript accordingly, and the reply content and revised manuscript have been added to the attachment, please check it.
yours sincerely,
Jinan Cheng

Reviewer 2 Report
The manuscript deals with a study conducted in an area in which Stellera chamaejasme is invasive. The authors report analyzes on the composition of the fungal and bacterial microbial communities, the chemical-physical properties of the soil, the enzymatic activities present and the allelopathic substances secreted by S. chamaejasme. The study is based on 16 samplings in four different degrees of plant cover, all at roughly the same height above sea level.
I suggest not dividing the abstract into Background, methods, results, and conclusions.
I have a note to make: the authors consider, report in the title and comment in the text on soil enzymes as independent soil activities detached from the microbial community without considering that they are enzymes almost exclusively produced by the same community and are therefore strictly dependent on it. Not the opposite. In particular, this must be reconsidered in several places starting with the title in which it is said that soil enzymes have an effect on the microbial community. Review the sentences on lines 78-80 and 84-85, for example, and in the discussion. Or in the sentence at lines 107-110 in which the hypothesis is advanced that there is a certain correlation between soil enzymes and the microbial community. Or in the sentence at lines 434-436 which I would reverse it.
The discussion has several repetitions and needs to be reviewed.
I suggest rephrasing the conclusions by expanding them and putting them in a broader context. As they are formulated, they lead to the conclusion that Stellera has a positive effect on the soil - and consequently on the environment - when it invades an area. But it is so?
Rephrase Material and Methods. For example you cannot say that “..the time was conducted..” (line 120) or “..the sample plots were repeated..” (lines 126-127).
It is not clear how you measured the enzymatic activities of the soil (lines 140-144).
Lines 21-22. “has not been clear” Please explain.
Lines 43-45. “It is also the most important place for rhizosphere microorganisms to multiply and grow, which contains rich microbial diversity, and is the main source of plant rhizosphere microorganisms” please rephrase.
Line 61: “Alliaria Fololata” please check and the f goes lowercase
Lines 64 and 361: Arabidopsis thaliana should be in italic.
Line 220. Change “soil acidity” into “soil pH”.
Line 258: Caption of Table 2: “Fungi” not “Fungus”.
Line 286: change “shows” with “is shown”.
Line 305: “Synthesize the above” correct to “To synthesize the above”.
Line 330: Change to “Redundancy analysis (RDA)”.
Figure 6: it is difficult to interpret all the acronyms mentioned in lines 336-345. Report them at least in the caption of the figure.
Author Response

(The authors gave the same response as above.)

Round 2
Reviewer 2 Report
The authors reviewed the manuscript following the recommendations. There are still some minor revisions of English to be done, I suggest one last revision by a native speaker.
Line 416: "Vivanco" add "et al."
Line 459: "In the soil that invaded by S. chamaejasme" change to "In the soil invaded by S. chamaejasme".
Author Response
Dear editor,
We have revised the reviewer's comments, and the revised content has been added to the attachment, please check.
Yours sincerely,
Jinan Cheng
